# Extremal Contours: Gradient-driven contours for compact visual attribution

Reza Karimzadeh[*1], Albert Alonso[*1], Frans Zdyb[1], Julius B. Kirkegaard[1], and Bulat Ibragimov[1]

[1]Department of Computer Science (DIKU), University of Copenhagen, 2100 Copenhagen, Denmark
{reza.karimzadeh, aaf, frzd, juki, bulat}@di.ku.dk

## Abstract

Faithful yet compact explanations for vision models remain a challenge, as commonly used dense perturbation masks are often fragmented and overfitted, needing careful post-processing. Here, we present a training-free explanation method that replaces dense masks with smooth tunable contours. A star-convex region is parameterized by a truncated Fourier series and optimized under an extremal preserve/delete objective using the classifier gradients. The approach guarantees a single, simply connected mask, cuts the number of free parameters by orders of magnitude, and yields stable boundary updates without cleanup. Restricting solutions to low-dimensional, smooth contours makes the method robust to adversarial masking artifacts. On ImageNet classifiers, it matches the extremal fidelity of dense masks while producing compact, interpretable regions with improved run-to-run consistency. Explicit area control also enables importance contour maps, yielding a transparent fidelity–area profiles. Finally, we extend the approach to multi-contour and show how it can localize multiple objects within the same framework. Across benchmarks, the method achieves higher relevance mass and lower complexity than gradient and perturbation based baselines, with especially strong gains on self-supervised DINO models where it improves relevance mass by over 15% and maintains positive faithfulness correlations.

## 1 Introduction

In the last decade, deep neural networks have consistently represented the state-of-the-art in the computer vision field, achieving strong performance in classification, detection, and segmentation tasks [1–4]. As these models are increasingly deployed in sensitive domains such as medical imaging [5, 6] and autonomous driving [7], among others, interpretability is needed to establish trust, diagnose errors, and ensure reliability [8, 9].

A family of explanation techniques are saliency maps, which attribute importance scores to individual input pixels or regions [10–12]. Gradient-based methods, such as saliency backpropagation and integrated gradients [13], visualize local sensitivities of the prediction with respect to the input. While computationally efficient, these approaches often highlight many, diffuse regions, are sensitive to noise, and may fail sanity checks that test for faithfulness [14].

An alternative line of work uses perturbation-based explanations, which measure how predictions change when parts of the input are masked or altered. By optimizing perturbations, methods such as Meaningful Perturbations [15] and Extremal Perturbations [16] identify regions that are most responsible for the output of a model. Perturbation approaches are more closely tied to causal influence [17], but typically rely on dense masks obtained through the gradients [16] or by training an auxiliary network [9]. A natural requirement for explanations is the ability to highlight compact, coherent regions that are sufficient to preserve or suppress a model prediction [18]. Such regions are easier to interpret, compare across inputs, and analyze quantitatively, but because of their lack of topological guarantees, masking methods are susceptible to noisy, fragmented, or multi-component outputs and require strong regularization [16, 19, 20]. Recent work has attempted to impose continuity and structural constraints on explanation masks, for example by learning implicit neural representations that generate smooth, area-conditioned masks [21]. While such formulations improve continuity, they still lack explicit geometric control and topological guarantees.

This work addresses these limitations by proposing a structured representation for perturbation masks based on gradient-driven contours. Instead of optimizing every pixel of the mask, we parameterize a closed star-convex region using a truncated Fourier series that defines the radial extent of the mask relative to a learnable center. Such geometry-aware parameterizations echo ideas from computational geometry, where analytic formulations like surface-patch Voronoi diagrams ensure smooth and topologically consistent boundaries [22]. This compact representation guarantees smooth, simply connected masks by construction and can be optimized end-to-end through any differentiable criterion, similar to how differentiable rendering is used to refine detec-

---

*These authors contributed equally.
Code available at https://github.com/rezakarimzadeh/extremal-contours

Proceedings of the 7th Northern Lights Deep Learning Conference (NLDL), PMLR 307, 2026.

tions [23].

Compared to existing extremal methods [16], our approach reduces the dimensionality of learnable parameters by one to two orders of magnitude and converges reliably without dataset-level optimization. The result is a concise, topology-preserving explanation that retains the faithfulness of perturbation-based approaches while avoiding the instability and complexity of learnable pixel masks.

## 2   Method

We represent explanations as smooth star-convex masks optimized under a perturbation objective. Each region is parameterized relative to a learnable center location $c \in \mathbb{R}^2$ by a truncated Fourier expansion,

$$\hat{r}(\theta) = r_0 + \Re_+\left(\sum_{k=1}^{K} w_k e^{ik\theta}\right), \qquad (1)$$

with complex coefficients $w_k \in \mathbb{C}$, yielding closed, smooth contours from only $2K+3$ free parameters. The operator $\Re_+$ takes the real part normalized to the positive range.

For a pixel $p = (x, y)$ in polar coordinates relative to $c$, with angle $\theta_p$ and radius $\rho_p$, we define its mask value as

$$m(p) = \frac{1}{1 + \exp\left(\tau \cdot [\hat{r}(\theta_p) - \rho_p]\right)}, \qquad (2)$$

where $\tau$ controls boundary sharpness.

Perturbations follow the extremal principle [15, 16]. A blurred background $\tilde{x}$ is produced by Gaussian smoothing of the input $x$, and the mask defines preserved and deleted variants,

$$x_{\mathrm{p}} = m \odot x + (1 - m) \odot \tilde{x}, \qquad (3)$$
$$x_{\mathrm{d}} = (1 - m) \odot x + m \odot \tilde{x}. \qquad (4)$$

The loss we minimize is the sum of three term: (1) the extremal loss, (2) a term that prefers smaller areas, and (3) a shape regularizer,

$$\mathcal{L} = \mathcal{L}_{\mathrm{extremal}} + \lambda_a \alpha_r + \lambda_r \mathcal{L}_{\mathrm{spec}}. \qquad (5)$$

Similar to Møller et al. [20], we construct the loss to encourage preserved regions to retain the model feature embedding, while deletion suppresses it:

$$\mathcal{L}_{\mathrm{extremal}} = -\cos(e_p, e_o) + \cos(e_d, e_o). \qquad (6)$$

Here, $e_o, e_p, e_d$ are the embeddings of the classifier backbone of the original image, the preserved and the deleted variants.

Since Eq. (6) benefits from large masked areas, we regularize the solution contour by its area fraction given by

$$\alpha_r = \frac{1}{2 \cdot S} \int_0^{2\pi} \hat{r}(\theta)^2 \, d\theta, \qquad (7)$$

normalized to the $[-1, 1]^2$ image domain ($S=4$). The hyperparameter $\lambda_a$ controls how much a decrease is loss is valued compared to an increase in area as $\partial_a \mathcal{L}_{\mathrm{extremal}} \approx -\lambda_a$ at the minimum. This is a user preference. In our experiments, we found that tuning it dynamically by

$$\lambda_a = \min(5, \frac{1}{1 - \cos(e_o, e_p)}) \qquad (8)$$

gives good results across image types. This schedule increases when the preserved embedding diverges from the original, ensuring that compactness is only enforced once fidelity is maintained. At $\cos(e_o, e_p) = 0$, the method is forced to identify a loss for which $\partial_a \mathcal{L}_{\mathrm{extremal}} \approx -1$, which is guaranteed possible by the mean value theorem. We do not propagate gradients through Eq. (8), which only serves to balance the contribution of the two loss terms.

Finally, since we typically prefer rounded, smoother shapes, high-frequency oscillations are discouraged by penalizing Fourier energy,

$$\mathcal{L}_{\mathrm{spec}} = \sum_{k=1}^{K} k^2 |w_k|^2, \qquad (9)$$

which suppresses unstable boundaries.

Optimization uses adaptive gradient steps [27] and $\tau$ annealing for improved convergence (see Appendix A and Algorithm 1 for more practical details).

## 3   Results

We evaluate our extremal contour masks using two pretrained classifiers: a supervised ResNet-50 [2] and a self-supervised DINO ViT-B/16 [24]. Our evaluation considers three complementary perspectives: the qualitative appearance of the explanations, their quantitative explainability, and the robustness of the optimization process. Finally, we explore the ability of the method to extend to images containing multiple objects.

### 3.1   Experimental Setup

Experiments were performed on two subsets: 100 ImageNet [1] validation images containing single objects and 100 COCO [28] images. For both datasets, images were paired with their bounding boxes or segmentation masks, and the subsets were fixed across all methods to ensure comparability. Following prior work [29], multiple annotations per image were merged into a single mask or bounding box to allow consistent metric computation.

To evaluate our method, we used established XAI metrics [30] grouped into three categories. The first

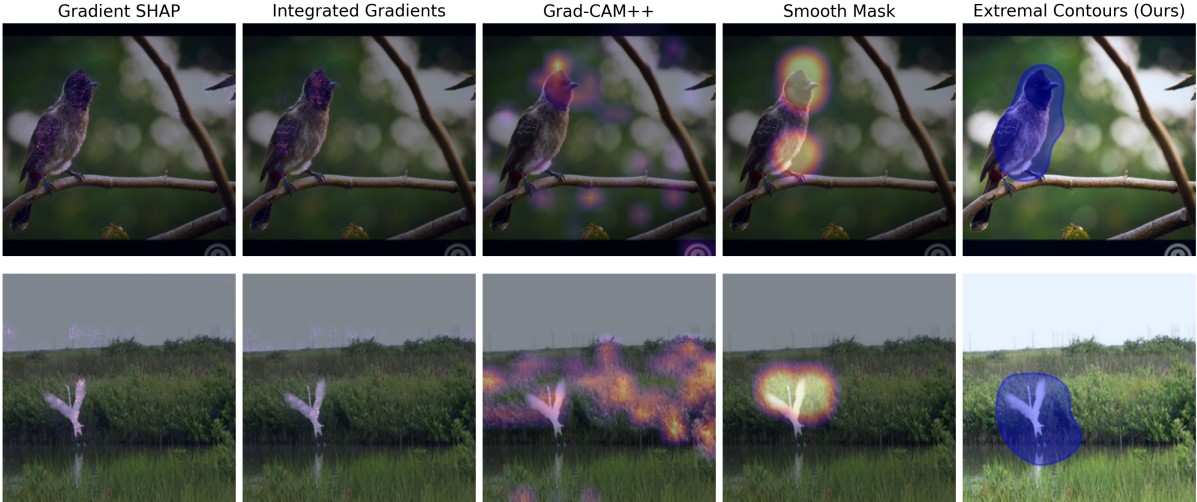

| Gradient SHAP | Integrated Gradients | Grad-CAM++ | Smooth Mask | Extremal Contours (Ours) |

**Figure 1.** Comparison of explanation methods on ImageNet validation images for the DINO [24] model. While gradient-based maps (e.g., Gradient SHAP [25], Integrated Gradients [13], Grad-CAM++ [12, 26]) and dense perturbation masks (Smooth Mask [16]) typically produce diffuse or sometimes fragmented attributions, our parameterization yields a single smooth, simply connected contour (Extremal Contour) that encloses the object of interest, highlighting a different representational paradigm for explainability.

---

**Algorithm 1** Extremal Contours Optimization

---

1: **Input:** image $x$, pretrained model $f$, mask parameters $\Theta = (c, r_0, w_1, \ldots, w_K)$
2: Initialize parameters $\Theta$ and the `AdamW` optimizer [27].
3: **for** each iteration $t = 1 \ldots T$ **do**
4:     Generate mask $m$ from Fourier radius (Eq. 1–2) with $\tau(t)$ annealing.
5:     Construct perturbed inputs $x_{\mathrm{p}}, x_{\mathrm{d}}$ (Eq. 3–4).
6:     Extract embeddings $e_o, e_p, e_d \leftarrow f(x), f(x_{\mathrm{p}}), f(x_{\mathrm{d}})$.
7:     Compute extremal loss $\mathcal{L}_{\mathrm{extremal}}$ (Eq. 6).
8:     Compute area penalty $\alpha_r$ with adaptive weight $\lambda_a$ (Eq. 8) and spectral penalty $\mathcal{L}_{\mathrm{spec}}$ (Eq. 9).
9:     Update $\Theta$ using `AdamW` on the total loss $\mathcal{L}$ (Eq. 5).
10:     **if** the loss $\mathcal{L}$ has not decreased for $P$ consecutive iterations **then**
11:         **break** {early stopping}
12:     **end if**
13: **end for**
14: **Output:** Optimized contour parameters $\Theta$.

---

is locality, which measures spatial agreement between an explanation and ground-truth annotations. We report relevance rank accuracy (RKA), the fraction of top-$k$ important pixels (with $k$ equal to the mask size) lying inside the annotation, and relevance mass accuracy (RMA), the ratio of positive attribution within the annotation to the total attribution mass. Higher values indicate stronger alignment with human labels [29, 31, 32].

The second category is complexity, which quantifies interpretability by sparsity. We use (i) complexity, defined as the entropy of the attribution distribution, and (ii) sparseness, measured as the Gini index of absolute attributions. Low entropy or high Gini indicate focused explanations [33, 34].

Finally, faithfulness assesses whether explanations truly reflect model reasoning. It is measured by perturbing or removing highly attributed regions and observing the prediction drop; greater decreases imply more faithful explanations [32, 35].

## 3.2 Qualitative Results

Figure 1 presents a comparison of explanation methods on ImageNet validation images for the DINO model. Gradient SHAP [25], Integrated Gradients [13], Grad-CAM++ [12, 26], and Smooth Mask [16] often produce diffuse, fragmented, or irregular attributions. In contrast, our extremal contour parameterization generates smooth and compact boundaries that consistently enclose the object of interest. This shift from pixel-level heatmaps to contour-based explanations provides a more structured and interpretable representation of model reasoning.

To illustrate the underlying perturbation objec-

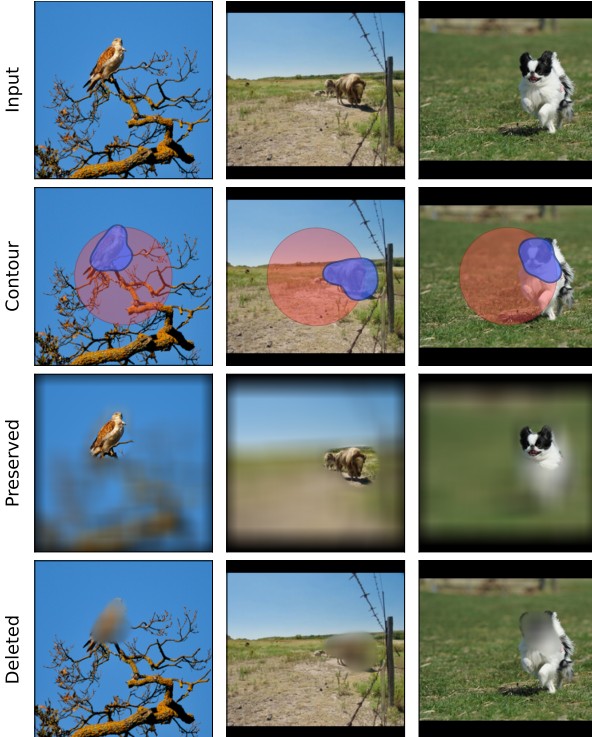

**Figure 2.** Qualitative results on ImageNet images. Each column: input image, our optimized mask (red: initial contour, blue: optimized contour), preserve variant, and deletion variant. Our method highlights compact star-convex regions that preserve predictions while their deletion strongly suppresses them.

tive, Figure 2 presents qualitative examples showing the input, the optimized mask, and the resulting preserved and deleted variants. These demonstrate that the selected region is sufficient to maintain the prediction, while its removal attempts to suppress it, confirming that the learned contours faithfully capture the features driving the model output.

## 3.3 Quantitative Results

Across both COCO and ImageNet benchmarks, the proposed extremal contour approach achieves competitive or superior performance compared to standard attribution methods. On COCO (Table 1), our method consistently attains the best relevance rank and mass, indicating stronger alignment with ground-truth object regions, while maintaining low complexity and high sparseness. On ImageNet (Table 2), Smooth Mask achieves the highest overall scores in most supervised settings, particularly in relevance-based metrics, but our method provides a favorable balance between localization fidelity and explanation compactness. Importantly, extremal contours show improved robustness in the self-supervised DINO model, outperforming other methods in relevance mass and complexity, while delivering positive faithfulness correlations where

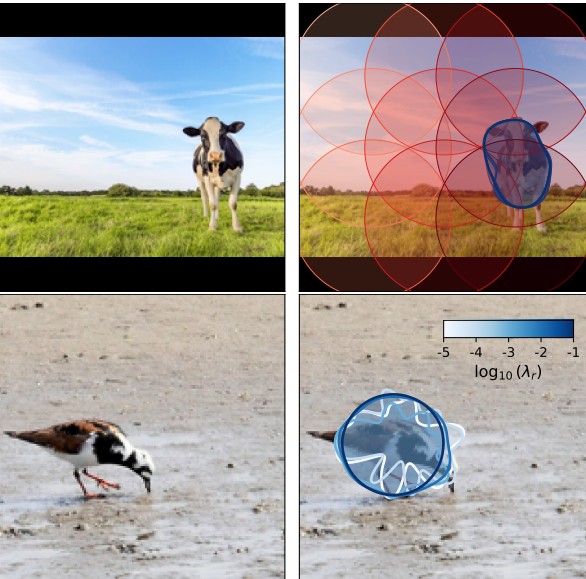

**Figure 3.** Robustness of the method. (Top) Red circles denote different initial positions $c$ of the contour, while the blue contour is the final optimized masks, overlapped. (Bottom) Effect of the spectral regularizer on contour complexity (color coded). Large $\lambda_r$ enforces smooth, near-circular masks, while lower values permit higher-frequency modes, yet result in the same location. Each trajectory is optimized independently, though we show them simultaneously for visualization.

most baselines fail. These results highlight that contour-based explanations not only capture object boundaries more reliably but also provide more stable and interpretable attributions across supervised and self-supervised models.

## 3.4 Robustness

Our formulation has only two major free choices: the initial contour center $c$ and the spectral regularization weight $\lambda_r$. To test sensitivity to $c$, we initialize contours from nine different starting points spread across the image (Fig. 3). In all cases, optimization converges to the same object with only minor variation in boundary shape, indicating that the method is stable and not reliant on initialization. We note, however, that the closer the initial contour is to the primary evidence region, the faster the convergence and the less prone it is to local optima, particularly when the object has irregular shapes.

The second parameter, $\lambda_r$, controls the degree of allowable high-frequency oscillations. In practice, we also limit the number of Fourier coefficients $K$ for computational efficiency. Figure 3 shows how large $\lambda_r$ yields smooth, near-circular contours, while smaller values allow more irregular boundaries. Despite these differences in presentation, all runs independently recover the same target object, demonstrating robustness to this regularization setting.

**Table 1.** Quantitative comparison of attribution methods on COCO validation images for a supervised ResNet-50 and a self-supervised DINO ViT-B/16. Extremal contours (ours) achieve strong localization performance while maintaining compact and concentrated explanations. Values are reported as mean (95% CI).

| Model | Method | Relevance Rank ↑ | Relevance Mass ↑ | Complexity ↓ | Sparseness ↑ |
|---|---|---|---|---|---|
| Supervised | Gradient SHAP [25] | 0.430 (0.373, 0.487) | 0.418 (0.361, 0.476) | 10.145 (10.107, 10.184) | 0.594 (0.582, 0.607) |
| | Integrated Grads [13] | 0.432 (0.375, 0.488) | 0.423 (0.365, 0.481) | 10.165 (10.118, 10.212) | 0.584 (0.570, 0.599) |
| | Smooth Mask [16] | 0.462 (0.406, 0.518) | 0.514 (0.443, 0.585) | **8.655** (8.630, 8.679) | **0.910** (0.908, 0.912) |
| | Grad-CAM++ [12, 26] | 0.460 (0.401, 0.519) | 0.465 (0.393, 0.537) | 9.518 (9.385, 9.651) | 0.708 (0.664, 0.751) |
| | **Extremal Contour** | **0.478** (0.427, 0.530) | **0.602** (0.537, 0.666) | 8.990 (8.929, 9.051) | 0.843 (0.834, 0.852) |
| DINO | Gradient SHAP [25] | 0.492 (0.438, 0.545) | 0.485 (0.428, 0.542) | 10.202 (10.164, 10.240) | 0.573 (0.560, 0.586) |
| | Integrated Grads [13] | 0.492 (0.438, 0.545) | 0.484 (0.428, 0.541) | 10.239 (10.210, 10.267) | 0.561 (0.550, 0.572) |
| | Smooth Mask [16] | 0.454 (0.398, 0.510) | 0.538 (0.469, 0.608) | **8.658** (8.630, 8.685) | **0.910** (0.908, 0.912) |
| | Grad-CAM++ [12, 26] | 0.434 (0.375, 0.493) | 0.432 (0.371, 0.494) | 9.993 (9.932, 10.054) | 0.651 (0.632, 0.670) |
| | **Extremal Contour** | **0.481** (0.429, 0.533) | **0.652** (0.586, 0.718) | 8.665 (8.616, 8.715) | 0.889 (0.884, 0.895) |

**Table 2.** Quantitative comparison of attribution methods on ImageNet validation images for a supervised ResNet-50 and a self-supervised DINO ViT-B/16. Extremal contours (ours) deliver competitive localization and simplicity while improving robustness and consistency relative to gradient- and perturbation-based methods. Values are reported as mean (95% CI).

| Model | Method | Relevance Rank ↑ | Relevance Mass ↑ | Complexity ↓ | Sparseness ↑ | Faithfulness ↑ |
|---|---|---|---|---|---|---|
| Supervised | Gradient SHAP [25] | 0.606 (0.546, 0.666) | 0.628 (0.563, 0.692) | 10.115 (10.066, 10.165) | 0.604 (0.588, 0.619) | 0.036 (-0.027, 0.099) |
| | Integrated Grads [13] | 0.614 (0.555, 0.673) | 0.632 (0.568, 0.696) | 10.163 (10.123, 10.202) | 0.589 (0.575, 0.603) | 0.070 (0.004, 0.137) |
| | Smooth Mask [16] | **0.638** (0.580, 0.696) | **0.806** (0.738, 0.874) | 8.686 (8.659, 8.712) | **0.907** (0.905, 0.910) | **0.091** (0.029, 0.153) |
| | Grad-CAM++ [12, 26] | 0.588 (0.524, 0.652) | 0.626 (0.552, 0.701) | 9.737 (9.597, 9.877) | 0.658 (0.614, 0.701) | 0.090 (0.018, 0.161) |
| | **Extremal Contour** | 0.596 (0.536, 0.655) | 0.779 (0.706, 0.852) | 8.878 (8.795, 8.961) | 0.855 (0.843, 0.866) | 0.050 (-0.022, 0.122) |
| DINO | Gradient SHAP [25] | 0.610 (0.549, 0.670) | 0.633 (0.568, 0.699) | 10.152 (10.096, 10.207) | 0.589 (0.573, 0.604) | -0.050 (-0.143, 0.042) |
| | Integrated Grads [13] | 0.610 (0.550, 0.669) | 0.637 (0.573, 0.701) | 10.210 (10.174, 10.246) | 0.571 (0.558, 0.584) | -0.000 (-0.096, 0.095) |
| | Smooth Mask [16] | **0.616** (0.555, 0.676) | 0.758 (0.683, 0.833) | 8.659 (8.636, 8.682) | **0.910** (0.908, 0.912) | 0.010 (-0.060, 0.081) |
| | Grad-CAM++ [12, 26] | 0.551 (0.486, 0.617) | 0.584 (0.513, 0.654) | 10.022 (9.959, 10.086) | 0.636 (0.613, 0.659) | -0.041 (-0.124, 0.042) |
| | **Extremal Contour** | 0.601 (0.544, 0.659) | **0.814** (0.745, 0.883) | **8.562** (8.504, 8.621) | 0.899 (0.894, 0.905) | -0.033 (-0.118, 0.051) |

In contrast, methods that learn dense masks directly can lead to adversarial solutions. The optimizer can exploit unconstrained degrees of freedom to satisfy the extremal loss (Eq. 6) without producing meaningful explanations. For instance, to migrate this, Fong et al. [16] and Møller et al. [20] rely on gaussian smoothing of lower dimensional masks, at the cost of fidelity. Our contour parameterization avoids this issue by construction. Our method is limited to select a compact, connected region, ensuring that explanations remain interpretable without post-hoc corrections.

## 3.5 Fixed-area explanations

Perturbation-based explanations often involve a trade-off between attribution area and faithfulness. In our formulation, area is controlled by the adaptive weight $\lambda_a$ described in Eq. (8), which automatically scales to enforce the smallest region that still preserves the embedding. This yields compact masks without requiring manual tuning.

Dense explanation methods often constrain to a fixed area size [16]. To explore this for our approach,

we can replace the adaptive area term with an objective that has a target fraction $\alpha^*$:

$$\mathcal{L} = \mathcal{L}_{\text{extremal}} + \lambda_a |\alpha_{\mathbf{r}} - \alpha^*| + \lambda_k \mathcal{L}_{\text{spec}}. \quad (10)$$

By varying $\alpha^*$, we can probe how mask extent influences embedding preservation. As shown in Fig. 4, tuning $\alpha^*$ produces contours of different sizes that remain optimized for attribution. The resulting collection of contours resemblance a contour-map of faithfulness, where successive closed curves highlight regions sufficient for the model prediction. As expected, the embedding preservation (deletion) is maximized (minimized) at large mask areas, whereas aiming for the smaller evidence results in a cost in performance.

The single-contour construction naturally suits images with one dominant object. Extending it to multi-object settings would require additional constraints to separate regions (see Sec. 3.6).

Finally, sweeping $\alpha^*$ recovers the characteristic area–faithfulness trade-off curve (Fig. 4), serving as a sanity check that the learned masks are placed in meaningful locations.

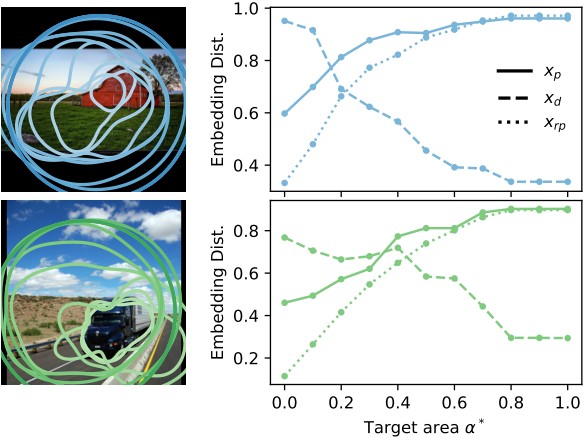

**Figure 4.** Area-fidelity trade-off. (Left) Single closed contours at target areas $\alpha^* \in \{0.1, \ldots, 0.7\}$ (small to large). The combined contours resemble a contour map of the faithfulness based on the available region of the image. (Right) Target class probability as a function of the targeted area $\alpha^*$. Solid lines shows the preserved variants whereas dashed lines show the deletion. Dotted lines show the average embedding preservation of randomly sampled circular masks.

## 3.6 Multiple Contours

While we have presented the method for single star-convex regions, many images contain multiple objects that prediction networks focus on. Therefore, we extend our formulation to allow several independent contours to be optimized simultaneously. Each contour mask $m_i$ is computed using Eq. (2), and the final composed mask is trivially obtained by the pixel-wise maximum:

$$\bar{m}(p) = \max_{i=1\ldots N} m_i(p). \tag{11}$$

This preserves differentiability while ensuring that the overall mask works as expected by the method.

The loss is extended by summing the relative ares $\alpha_r^{(i)}$ and spectral penalties of each contour. Note that since areas are computed directly on the shape, the method encourages the contours to remain compact and discourages from overlapping, as opposed to the classical approach of counting mask pixels. In practice, the only initialization constraint we observe is due to complete overlap of the contours, where the gradient information is the same.

Figure 5 shows examples with $N=2$ and $N=4$ contours applied to multiple images with more than one element leading the prediction. This also shows the robustness of the method to consistently land on the objects that lead the decision making of the classifier. This demonstrates that our presented gradient-driven contour method can be extended to multi-object without sacrificing stability or interpretability. Nevertheless, a limitation of the presented formulation is its isotropic bias, which favors

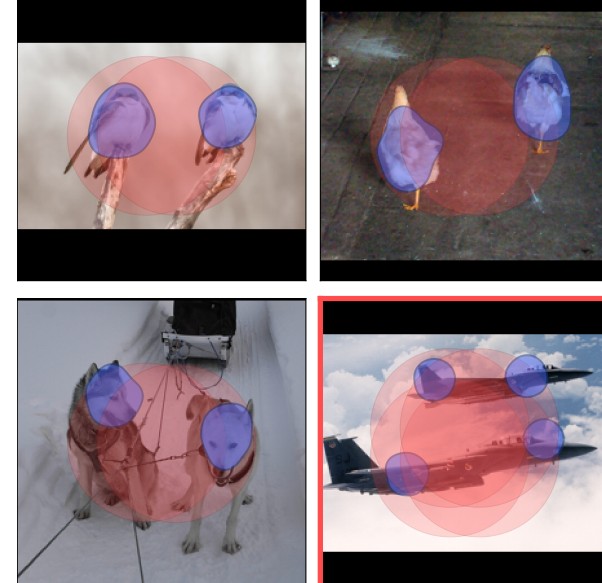

**Figure 5.** Examples of multiple contour optimization with $N=2$ (top, left/right; bottom left) and $N=4$ (bottom right). Optimized contours (blue) adapt to distinct salient objects within the image from the initial contours (red). The method encircles the regions that lead the classification. (bottom right) shows a failure case where the contours are not able to cover the salient objects in the image.

rounded, compact shapes. This results in elongated objects not being well captured even when multiple contours are optimized jointly.

## 4 Discussion

We proposed a Fourier parameterization for perturbation masks that produces smooth, simply connected regions and converges reliably under gradient descent. Compared to dense extremal masks [16, 36], our approach achieves similar fidelity while guaranteeing compact, interpretable shapes. This makes it useful in settings where stability and topological consistency result in higher explainability than direct pixel attribution [14].

The main advantage of our method is simplicity. With only a small set of parameters, optimization is direct and reproducible, avoiding the instability seen in less constrained methods [16, 19]. Similarly, the Fourier basis also allows explicit control over smoothness and contour complexity through a single regularization term.

Nevertheless, the method has a few limitations. The star-convex constraint ensures that every point in the mask is directly visible from its center, which guarantees smooth, single-component regions but prevents capturing objects with strong concavities or holes. Because the masks enclose contiguous areas, they may also include non-informative pixels, which

lowers sparsity and complexity compared to saliency maps [12, 13, 37]. In addition, unlike methods that produce attribution maps, our approach results in a binary mask, which can reduce the granularity of explanations. For fine-grained classes, the reduced flexibility under performs dense masks. Finally, optimization requires iterative updates of the contour rather than a single backward pass, so runtime is higher and efficiency remains an open direction.

We also showed that the formulation extends naturally to multiple contours, allowing disjoint regions of evidence. Beyond ImageNet classifiers, a natural next step is to deploy contour-based explanations in domains where compact, clinician-friendly attributions matter, most notably medical imaging (CT, MRI, pathology slides) settings [5, 6, 38]. In these settings, learned contours can serve as editable suggestion masks that radiologists refine with minimal effort, thereby reducing the annotation burden compared to dense pixel-wise labeling while maintaining faithfulness. More broadly, such compact and didactic explanations can support training and assessment workflows, improve reader consistency and evaluation quality, and naturally avoid fragmented voxel islands within a single object segmentation [39].

The proposed method is inherently model-agnostic and can be readily extended to other vision tasks. For object detection, the backbone continues to produce embeddings and target scores, enabling contour optimization with respect to a chosen box or query score using unaltered gradients. For segmentation, the extension is more intricate and warrants investigation into which loss formulations yield the most informative and faithful attributions. Owing to its simplicity, the framework allows users to flexibly select losses that align with their analytical objectives.

These directions warrant systematic exploration in future studies, including the development of richer contour parameterizations that balance expressivity with topological guarantees, enabling multi-component or hierarchical masks while preserving the efficiency and stability of the Fourier formulation.

## Acknowledgments

We thank Bjørn L. Møller for helpful suggestions and feedback, and Madeleine Wyburd and Malte Silbernagel for engaging discussions. This work was supported by the Novo Nordisk Foundation, Denmark under Grant NFF20OC0062056.

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

# A   Implementation Details

The optimization process involves a few practical considerations that make the method stable and reproducible.

**Initialization.**   The contour center $c$ is placed at the image center for simplicity (any location is acceptable as shown in Sec. 3.4). Fourier coefficients initialized to zero ($\omega_k = 0, \forall k \in (0, K)$) and $\tau=1$, the contour reduces to a circle of radius $r_0=0.5$ in the normalized $[-1, 1]^2$ domain. This initial mask alongside its smooth boundary cover (gradient range) most of the image, ensuring that gradients are available everywhere so the optimizer can relocate $c$ if needed.

**Radius and Mask.**   Each region is parameterized relative to $c$ by a truncated Fourier expansion with bounded radial deviations:

$$\hat{r}(\theta) = r_0 + \bar{s} \tanh\left(\text{Re} \sum_{k=1}^{K} \omega_k e^{ik\theta}\right), \quad (12)$$

where $\bar{s}$ is a scaling factor given by $\bar{s} \equiv \min(r_0 - R_{min}, R_{max} - r_0)$ with $R_{min}=0.1$ and $R_{max}=1.0$.

**Frequency budget and regularization.**   In the experiments shown here, we set $K=5$, which already yields expressive and rounded masks while keeping the parameterization compact. Larger $K$ values are supported, but the spectral regularizer $\mathcal{L}_{spec}$ naturally suppresses high-frequency coefficients so that unused harmonics decay during optimization. In Sec. 3.4 we show results with $K=20$ and varying $\lambda_k$, illustrating how the choice of frequency budget and regularization strength affects the mask shape.

The original embedding $e_o = f(x)$ and the blurred background image $\tilde{x}$ are both computed once before the optimization loop, since they remain fixed throughout. Cosine similarity is evaluated as

$$\cos(A, B) = \frac{A^\top B}{\|A\| \|B\|}, \quad (13)$$

without separately normalizing the embeddings.

The blurred background $\tilde{x}$ is obtained using Gaussian smoothing with kernel size 21 and $\sigma=20$. We observed that the method is robust to moderate changes of these parameters (e.g. $\sigma \in [10, 30]$). Unlike the "soft mask" variant of extremal perturbations [15, 16], we keep the blur scale fixed, which simplifies optimization and avoids introducing mask-dependent artifacts.

**Sharpness annealing**   During optimization, the sharpness parameter $\tau$ is annealed according to a cosine schedule:

$$\tau(t) = \tau_0 + \tfrac{1}{2} (\tau_\infty - \tau_0) \left[1 - \cos\left(\pi \tfrac{t}{T}\right)\right], \quad (14)$$

with $\tau_0=1$, $\tau_\infty=100$, and $T$ the total number of iterations. This schedule yields smooth gradients in early iterations and nearly binary masks at convergence. This is a trick to deal with the information range during the gradient calculations. Note also that solutions tends to converge before reaching $T$, hence we also add a convergences early stopping for efficiency.

**Area schedule**   The adaptive area weight $\lambda_a$ is clipped to an upper bound of 5.0 to balance the loss terms, since $\mathcal{L}_{extremal}$ is bounded within $[-2, 2]$. Gradients are stopped before computing $\lambda_a$ to prevent it from interfering with the parameter updates.

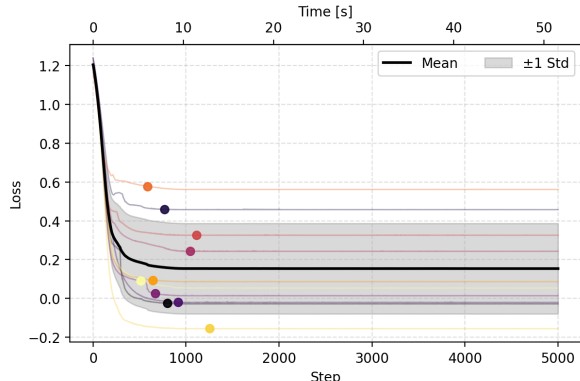

**Figure B.1.** Convergence behavior of the Extremal Contour method across 10 images. The runtime stabilizes around 1000 iterations, indicating that the stopping criteria effectively ensure contour stability. The dots indicate the iteration our methods assumes converges, around 8 seconds.

**Optimization** All parameters $(c, r_0, w_1, \ldots, w_K)$ are optimized with `AdamW` [27], using a learning rate of $\eta = 0.003$ and standard $\beta$ values. This combination provides stable convergence without the need for additional post-processing, and reliably produces smooth, star-convex masks across images.

# B    Runtime Analysis

To assess and compare the computational efficiency of different attribution methods, we measured their runtime on a set of 10 images and report the average and standard deviation (STD) values in Table B.1. All experiments were conducted on an NVIDIA GeForce RTX 3090 GPU. Owing to the iterative nature of the Extremal Contour algorithm, we further performed a convergence analysis over 10 independent images to examine its stability across iterations. As illustrated in Figure B.1, the optimization stabilizes after approximately 1000 iterations, indicating convergence to a consistent loss plateau. For completeness, we display the full convergence curves up to 5000 iterations, where the stopping points (marked as dots) demonstrate that all runs automatically terminated upon reaching the plateau according to the defined stopping criteria.

**Table B.1.** Average runtime (in seconds, mean ± std) for processing 10 images on NVIDIA GeForce RTX 3090.

| Method | Runtime (s) |
| --- | --- |
| Gradient SHAP | 0.032 ± 0.003 |
| Integrated Gradients | 0.099 ± 0.005 |
| Smooth Mask | 6.08 ± 0.04 |
| Grad-CAM++ | 0.020 ± 0.003 |
| Extremal Contour (ours) | 8.6 ± 2.5 |

