# OpenReview forum: "Extremal Contours: Gradient-driven contours for compact visual attribution"
_NLDL.org/2026/Conference — NLDL 2026 Oral_

### Official Review · Reviewer_VFKL · 2025-10-02
**Clear and solid paper introducing a new post-hoc explainability method.**

**Rating:** 5
**Confidence:** 3
**Final Rating:** 5
**Final Confidence:** 3

**Summary:**

The paper introduces a new post-hoc explainability method that creates a star-convex mask (possibly with smooth contours) with a few learnable parameters. The authors show both qualitative and quantitative results and comparisons to state-of-the-art methods. The method is sound and the results seem correct.

**Strengths:**

- The paper is easy to read.
- The results of the quantitative evaluations show that the new method is comparable to or better than the other methods tested (Gradient SHAP, Integrated Gradients, Smooth Mask, Grad-CAM++).
- The authors present multiple options for extensions or adjustments and comment on their advantages and disadvantages.
- I had some questions and uncertainties while reading the paper, but almost all of them got addressed sooner or later, so it seems to me that the authors thought everything through.

**Weaknesses:**

- There is a letter $a$ used in a subscript without explicitly defining what it means (around lines 126-139). I thought it was just a notation in $\lambda_a$, but then what does $\partial_a \mathcal{L}_{extremal}$ mean? There is no variable $a$ defined anywhere, as far as I can tell.

- There are no confidence intervals or uncertainties in the tables, so it is hard to estimate the significance of the results.

**Final Justification:**

The authors addressed all my remarks. Therefore, I maintain the scores as they are.

**Justification:**

I think the paper provides an interesting new method that provides some advantages over the existing methods. The authors provide a thorough evaluation of the method, both qualitatively and quantitatively. The paper is clearly written with few errors or ambiguities.

---

> ### Author Rebuttal · Authors · 2025-10-21
>
> Thank you for the positive and thorough review. We address the two questions below.
>
> **Notation around lines 126–139.** The subscript $a$ in $\partial_a \mathcal{L}$ refers to the area term. In our notation this is the area fraction $\alpha_r$. This is indeed a mistake.
> To avoid confusion, we write $\partial_{\alpha_r}\mathcal{L}$ and keep the symbols consistent with Eq. (7) and the adaptive weight $\lambda_\alpha$ defined in Eq. (8).
> This aligns the derivative, the area regularizer $\alpha_r$, and $\lambda_\alpha$ in that paragraph. Thank you for pointing it out.
>
> **Uncertainty reporting.** The tables now report mean values with 95% confidence intervals for all metrics on both COCO and ImageNet.
> We will make this explicit in the Captions and we hope now the results are better contextualized.
>
> Here are the included tables with the confidence intervals:
>
> | Model          | Method                | Relevance Rank ↑         | Relevance Mass ↑         | Complexity ↓             | Sparseness ↑             |
> | -------------- | --------------------- | ------------------------ | ------------------------ | ------------------------ | ------------------------ |
> | **Supervised** | Gradient SHAP [23]    | 0.430 (0.373, 0.487)     | 0.418 (0.361, 0.476)     | 10.145 (10.107, 10.184)  | 0.594 (0.582, 0.607)     |
> |                | Integrated Grads [13] | 0.432 (0.375, 0.488)     | 0.423 (0.365, 0.481)     | 10.165 (10.118, 10.212)  | 0.584 (0.570, 0.599)     |
> |                | Smooth Mask [16]      | 0.462 (0.406, 0.518)     | 0.514 (0.443, 0.585)     | 8.655 (8.630, 8.679)     | 0.910 (0.908, 0.912)     |
> |                | Grad-CAM++ [12, 24]   | 0.460 (0.401, 0.519)     | 0.465 (0.393, 0.537)     | 9.518 (9.385, 9.651)     | 0.708 (0.664, 0.751)     |
> |                | **Extremal Contour**  | **0.478 (0.427, 0.530)** | **0.602 (0.537, 0.666)** | **8.990 (8.929, 9.051)** | **0.843 (0.834, 0.852)** |
> | **DINO**       | Gradient SHAP [23]    | 0.492 (0.438, 0.545)     | 0.485 (0.428, 0.542)     | 10.202 (10.164, 10.240)  | 0.573 (0.560, 0.586)     |
> |                | Integrated Grads [13] | 0.492 (0.438, 0.545)     | 0.484 (0.428, 0.541)     | 10.239 (10.210, 10.267)  | 0.561 (0.550, 0.572)     |
> |                | Smooth Mask [16]      | 0.454 (0.398, 0.510)     | 0.538 (0.469, 0.608)     | 8.658 (8.630, 8.685)     | 0.910 (0.908, 0.912)     |
> |                | Grad-CAM++ [12, 24]   | 0.434 (0.375, 0.493)     | 0.432 (0.371, 0.494)     | 9.993 (9.932, 10.054)    | 0.651 (0.632, 0.670)     |
> |                | **Extremal Contour**  | **0.481 (0.429, 0.533)** | **0.652 (0.586, 0.718)** | **8.665 (8.616, 8.715)** | **0.889 (0.884, 0.895)** |
>
> | Model          | Method                | Relevance Rank ↑         | Relevance Mass ↑         | Complexity ↓             | Sparseness ↑             | Faithfulness ↑             |
> | -------------- | --------------------- | ------------------------ | ------------------------ | ------------------------ | ------------------------ | -------------------------- |
> | **Supervised** | Gradient SHAP [23]    | 0.606 (0.546, 0.666)     | 0.628 (0.563, 0.692)     | 10.115 (10.066, 10.165)  | 0.604 (0.588, 0.619)     | 0.036 (-0.027, 0.099)      |
> |                | Integrated Grads [13] | 0.614 (0.555, 0.672)     | 0.652 (0.586, 0.719)     | 10.136 (10.123, 10.202)  | 0.589 (0.575, 0.603)     | 0.070 (0.004, 0.137)       |
> |                | Smooth Mask [16]      | 0.638 (0.580, 0.696)     | 0.806 (0.738, 0.874)     | 8.686 (8.653, 8.712)     | 0.907 (0.905, 0.910)     | 0.090 (0.028, 0.153)       |
> |                | Grad-CAM++ [12, 24]   | 0.588 (0.524, 0.651)     | 0.622 (0.557, 0.701)     | 9.737 (9.597, 9.877)     | 0.658 (0.614, 0.701)     | 0.090 (0.018, 0.161)       |
> |                | **Extremal Contour**  | **0.596 (0.536, 0.655)** | **0.779 (0.720, 0.838)** | **8.855 (8.803, 8.866)** | **0.850 (0.820, 0.922)** | **0.050 (-0.020, 0.122)**  |
> | **DINO**       | Gradient SHAP [23]    | 0.610 (0.549, 0.670)     | 0.633 (0.568, 0.699)     | 10.152 (10.096, 10.207)  | 0.589 (0.573, 0.604)     | -0.050 (-0.143, 0.042)     |
> |                | Integrated Grads [13] | 0.610 (0.550, 0.669)     | 0.637 (0.573, 0.701)     | 10.210 (10.174, 10.246)  | 0.571 (0.558, 0.584)     | -0.090 (-0.096, 0.095)     |
> |                | Smooth Mask [16]      | 0.616 (0.556, 0.676)     | 0.804 (0.736, 0.882)     | 9.019 (8.986, 9.052)     | 0.910 (0.908, 0.912)     | 0.010 (-0.060, 0.080)      |
> |                | Grad-CAM++ [12, 24]   | 0.551 (0.486, 0.617)     | 0.584 (0.513, 0.656)     | 10.022 (9.959, 10.086)   | 0.636 (0.613, 0.659)     | -0.041 (-0.124, 0.042)     |
> |                | **Extremal Contour**  | **0.601 (0.544, 0.659)** | **0.814 (0.745, 0.882)** | **8.562 (8.504, 8.621)** | **0.899 (0.894, 0.905)** | **-0.033 (-0.118, 0.051)** |

---

### Official Review · Reviewer_uUNh · 2025-10-07
**Interesting approach with boundary adhering explanations**

**Rating:** 4
**Confidence:** 4
**Final Rating:** 4
**Final Confidence:** 4

**Summary:**

The paper introduces a gradient optimized contour representation for saliency maps for interpretability and XAI. The idea is interesting, novel, and intuitive.

The main idea introduces a Fourier-parametrization of a star-convex contour as a perturbation mask. This provides smooth, topologically connected contours that can be optimized with gradient search. The idea is that this provides a low-dimensional representation of connected areas of the image as units for counterfactual optimization, which have attractive properties, such compactness and connectivity.

**Strengths:**

1. The parametrization is novel, simple. and elegant, and has applications beyond explainability.
2. Positing the approach in a geometric framework provides a theoretical grounding that other saliency approaches often lack. This places the approach both in classic computer vision as well as within the context of deep learning.
3. Faithfulness is a grounded, well established principle with strong counterfactual grounding, which allows for directly quantifying the inherent trade off with compactness.
4. The method is post-hoc and relatively model agnostic, making the approach attractive in applied settings.
5. The method and evaluation is well argued, and comes across as robust. The authors use a well-balanced set of metrics for evaluation (but with some limitations on scope). Overall, the presentation is very well done.

**Weaknesses:**

1. Star convexity has some limitations with expressivity for highly irregular regions. This means the method cannot capture non-connected interpretations such as multiple objects. This can likely be solved by adding more regions, but the details are not expanded upon in the manuscript.
2. Gradient optimization requires multiple passes through the model. This is a limitation compared methods one-pass methods like GradCAM. This reviewer would argue that this is minor in general cases, but should be expanded upon in the manuscript.
3. The scope of evaluation is very minor; as this reviewer reads the results, only 200 images are included? This is not a large enough sample size to make the claims the authors make in the paper.
4. Comparisons that evaluate compactness benefit the approach, since it is a prerequisite assumption. Hence, this line of argument slightly misses the point in this reviewers opinion. Should saliency always be compact? Is it a strong metric for interpretable results?

**Final Justification:**

The work demonstrates a fresh approach using a contour based explainability method, with interesting general applications. The presentation is solid, and work is well written and clearly laid out. While the experimental results are slightly narrow, with a low sample size, the work is novel enough to be accepted to the conference.

**Justification:**

While the paper is limited in scope, the contributions are clear. There are several applications for the approach beyond the case of interpretability, and the paper argues convincingly for the cases where the approach has inherent benefits. Even if the method would have benefitted from more baselines, the idea is crystal clear, and the methodology is robust, even with a small sample size.

This reviewer believes the strengths outweighs the weaknesses, and the paper should be accepted to the conference.

---

> ### Author Rebuttal · Authors · 2025-10-21
>
> Thank you for the thoughtful and positive review. We address the four raised points below.
>
> **Expressivity and star convexity.** We agree that star convexity cannot capture some highly irregular shapes with one region. The choice was intentional: a low-dimensional, stable parameterization helps adoption by not overcomplicating the method's presentation and keeps the optimization well behaved. The method already supports multiple contours when needed which could solve harder to fit shapes. Similarly, there is nothing that technically blocks a fully free contour (for example, periodic spline control points). That path would require extra regularization and complexity, which risks diluting the core idea. In practice, the current parameterization covers most objects in our datasets; for rarer, more intricate cases, richer geometric priors can be used.
>
> **Gradient optimization cost.** We added a runtime section in the appendix, with a fixed hardware and protocol. As expected, our method is slower than one-pass saliency (about 8s per image) but similar in scale to other perturbation-based approaches (for example, Smooth Mask takes about 6s). Early stopping keeps runtimes usable. We also list straightforward engineering speedups, including feature caching, smaller Fourier budgets, and batched backbone calls. Since the optimization converges, a good initialization of the contour should reduce time further, and we plan to explore this in the future with a more application-focused objective.
>
> | **Method**               | **Runtime (s)**        |
> |---------------------------|------------------------|
> | Gradient SHAP             | 0.0318 ± 0.0025        |
> | Integrated Gradients      | 0.0986 ± 0.0055        |
> | Smooth Mask               | 6.0833 ± 0.0444        |
> | Grad-CAM++                | 0.0195 ± 0.0025        |
> | Extremal Contour (ours)   | 8.5740 ± 2.5312        |
>
>
> **Evaluation scope and sample size.** The study uses fixed subsets of ImageNet and COCO to enable controlled, repeatable comparisons across several metrics and models. We acknowledge the sample is modest. Expanding the evaluation is straightforward, but outside the rebuttal timeline. The reported trends are stable across runs and are presented with confidence intervals.
>
> **On compactness as a metric.** Compactness is not assumed to be universally desirable. Our goal is to study the trade-off between faithfulness and spatial simplicity when the user wants a clear region explaining a decision. The fixed-area sweep and area–faithfulness curve make this trade-off explicit. As the reviewer says, evidence can be non-compact. For those situations, multiple contours or richer priors are likely more appropriate.
>
> Your assessment very clearly highlights exactly what we aimed for: a simple, geometric, model-agnostic contour explanation that is easy to optimize and reason about. The expressivity–simplicity and speed–faithfulness trade-offs are transparent, and the method can be extended when applications call for more complex shapes or larger evaluations.

---

### Official Review · Reviewer_bwpZ · 2025-10-08

**Rating:** 2
**Confidence:** 4
**Final Rating:** 4
**Final Confidence:** 4

**Summary:**

This submission presents a training-free, gradient-driven visual attribution method for explaining vision models by parameterizing the explanation as a smooth, star-convex contour optimized under an extremal perturbation objective. The contour is represented using a truncated Fourier series, guaranteeing simply connected and interpretable regions—contrasting with fragmentary or dense pixel attributions from common baselines. Experiments are conducted on ImageNet and COCO. It shows that the method produces competitive localization, with higher sparsity and robustness, especially for self-supervised models. Extensions to multiple contours and profile-based importance mapping are also explored.

**Strengths:**

**(S1)** This work introduces a compact parameterization of explanation masks as smooth star-convex contours using Fourier basis. The framework optimizes these contours with respect to an extremal preservation/deletion objective,  ensures single-component interpretable regions, and avoids the fragmentation and adversarial artifacts typical in dense mask approaches.

**(S2)** The method is flexible, requiring orders of magnitude fewer parameters. Tab, 1 and 2 show competitive performance on both COCO and ImageNet datasets, and especially strong gains in relevance and sparsity on DINO self-supervised models. Figure 3 also shows the robustness to initialization and regularization hyperparameters. This indicates favorable practical stability, in which existing extremal methods are weak.

**(S3)** The extension to multiple contours (Sec. 3.6, Fig. 5) is logically consistent and empirically demonstrated, indicating the broader potential value of this work.

**(S4)** The manuscript is generally well-presented. Notably, Fig, 1 and 2 clearly illustrate the method’s qualitative advantages in producing clean, human-interpretable regions as compared to fragmented or noisy baselines. It also acknowledges limitations, such as handling concave or elongated shapes and efficiency trade-offs.

**Weaknesses:**

**(W1)** The method’s restriction to star-convex contours (Sec. 4) prevents capturing objects with strong concavities, holes, or extreme elongation. This isotropic bias is evident in failure cases in Fig. 5 (bottom right). It may affect applicability to real-world images where salient regions are multi-lobed or disjoint—potentially lowering both localization fidelity and faithfulness relative to more flexible pixel-wise or multi-component baselines.

**(W2)** As discussed in Sec. 4 and Tab. 2, the method yields primarily binary or low-granularity masks versus soft/hierarchical attributions in widely used methods like Grad-CAM++. For fine-grained or small object classes, this leads to a loss of nuance and can hurt interpretability and end-user trust.

**(W3)** I suppose it would be more representative to conduct evaluation on more challenging datasets or tasks (e.g., multi-instance segmentation or fine-grained classification) where the limitations in star-convex parameterization might be more severe. I encourage the authors to  conduct experiments on these challenging scenarios in the future.

**(W4)** The choice of the area penalty schedule $\lambda_a$ (Eq. 8, Sec. 2, Sec. 3.5) is described as empirical and not justified with clear theoretical analysis. It seems unclear whether this always yields Pareto-optimal mask-fidelity trade-offs, or could systematically under or over-regularize in hard examples. I recommend the authors include more theoretical analysis on this point.

**(W5)** The mask construction in Eq. 2 is described as “normalized to the positive range” with on explanation on how normalization is done for arbitrary $K$ and input images, which may affect reproducibility. IMHO, technical details such as differentiability of the “max over $N$ contours” (Sec. 3.6) and gradient flow for the composite mask should be further clarified.

**(W6)** The manuscript does not discuss recent closely related methods utilizing implicit neural representations, hierarchical mask decomposition, or contour-based visualizations listed below:

- *Generating Visual Explanations from Deep Networks Using Implicit Neural Representations*, WACV 2025. This work presents visual attribution methods using implicit neural representations, directly addressing mask compactness and smoothness. IMHO, it should be discussed in the related work section and referenced in direct empirical or conceptual comparison to strengthen the context and novelty claims.
- *Towards Voronoi Diagrams of Surface Patches*, TVCG 2025. It examines contour-based partitioning and is relevant to smooth/compact mask generation. Referencing it can help better explain the Fourier contour approach relative to geometric representations.

**(W7)** The method admits higher computational cost due to iterative mask optimization vs. single-pass explainers like Grad-CAM or saliency maps. Practical usability in time-sensitive applications is thereby unclear. For example, what is the runtime and memory overhead for the contour method compared to e.g., Soft Mask or Grad-CAM++? How severe is the computational bottleneck in practice for large-scale or real-time scenarios? I believe this would be valuable for the community.

**Final Justification:**

The rebuttal has addressed my most critical, practical concerns. The new runtime analysis is particularly helpful (addressing my W7); quantifying the cost and comparing it against Smooth Mask (~6s) clarifies the method's practical trade-offs.

The clarifications regarding mask normalisation (W5, mapping to [0, 1]) and the max composition, and the commitment to update the appendix and open-source the code, resolve my main reservations about reproducibility.

My other points concerning the inherent limitations of the star-convex prior (W1) and the lack of graded, pixel-level attribution (W2) remain. I accept the authors' position that these are intentional design choices, trading expressivity for simplicity and stability. While I stand by my evaluation that this may limits the method’s applicability for highly complex or non-convex objects, this paper's core contribution is sound.

Given that my primary technical and reproducibility concerns have been resolved, I raise my rating and recommend acceptance. I hope that all improvements and clarifications promised during the rebuttal would be integrated into the final camera-ready version.

**Justification:**

This submission proposes a training-free visual attribution method that parameterizes explanations as smooth, star-convex contours via a truncated Fourier series, optimized under an extremal perturbation objective. However, several critical shortcomings prevent acceptance in my view.

First, the star-convex constraint raises some unexamined concerns about expressivity. It seems cannot represent objects with concavities, holes, or extreme elongation (Fig. 5, bottom right), which are common in real-world imagery. This structural bias might compromise both localization fidelity and faithfulness in multi-part or complex scenes. In addition, it produces near-binary masks, lacking the soft, hierarchical attribution detail of widely used techniques like Grad-CAM++. This reduces interpretability for fine-grained tasks where nuanced relevance gradients matter.

Second, evaluation is limited to single-object ImageNet and COCO subsets. It avoids more challenging settings like fine-grained classification or multi-instance segmentation, where the star-convex assumption would be most stressed. IMHO, the claimed robustness remains unproven without such validation.

Third, key design choices lack theoretical grounding. The adaptive area penalty in Eq. 8 is heuristic, with no analysis of its optimality or failure modes. Similarly, the mask normalization in Eq. 2 ("normalized to the positive range") is under specified. This raises my concerns  on reproducibility. The multi-contour extension in Eq. 11 asserts differentiability of the max-composition but does not provide analysis of gradient competition or convergence under overlap. Moreover, the computational cost is nontrivial due to iterative optimization, yet no runtime or memory results are provided. This may affect practical deployment in time-sensitive applications.

In sum, the contour-based formulation improves stability in restricted settings. However, the aforementioned limitations fall short of the bar for acceptance. A substantial revision would be required for reconsideration. I hope these comments help to strengthen this paper further and help the authors, fellow reviewers, and ACs understand the basis of my recommendation. I am also open to follow-up discussions to reach a consensus for the final decision.

---

> ### Author Rebuttal · Authors · 2025-10-21
>
> Thank you for the careful read and for engaging with both the strengths and the limits of the approach. We also appreciate the related-work pointers and now include a brief note situating recent implicit-representation and contour/partitioning methods alongside our explicit Fourier-contour formulation.
>
>
> **On the contour prior.** We chose a star-convex, low-dimensional contour to keep the explanation simple, stable, and easy to optimize. This does constrain extreme concavities or holes in a single region. In practice, two mechanisms already in the paper mitigate this: multiple contours for disjoint or multi-lobed evidence, and a higher Fourier budget for more flexible boundaries. There is no technical barrier to a fully free contour (for example, periodic spline control points), but that introduces extra regularization and complexity that would shift the focus away from the central idea. On the ImageNet and COCO subsets we used, the present parameterization covered most objects; for rarer, intricate cases, richer geometric priors are appropriate.
>
> **On graded insight vs. binary masks.** We agree that nuanced attribution is useful. Our aim is a clear, connected region when a crisp summary matters. To provide gradation without losing topological coherence, Section 3.5 performs a fixed-area sweep that yields nested contours and an area–faithfulness curve, giving a hierarchical view at the region level. For small or fine-grained classes, multiple contours capture separate pockets of evidence. Dense heatmaps like Grad-CAM++ remain complementary when per-pixel gradation is the goal.
>
> **On scope.** We agree that the evaluation is modest, which we do to keep comparisons controlled across metrics and models. We fully agree that broader settings such as fine-grained classification and multi-instance segmentation would stress the assumptions in interesting ways. Expanding the study is a natural next step that we aim to explore in future work, but beyond the current rebuttal window.
>
> **On schedules and implementation details.** Thanks for flagging clarity here. The area schedule is empirical and functions as a continuation scheme: it discourages trivial collapse or unchecked expansion, letting the contour expand when evidence is far and contract when close. Section 3.5 already shows this via a fixed-area sweep, which in practice is equivalent to scanning $\lambda_\alpha$. There is, in-practice, nothing preventing other schedules for $\lambda_\alpha$ in the framework.
>
> Regarding reproducibility, by “normalized to the positive range” we mean that the contour rasterization is mapped to a smooth mask in $[0,1]$ so the perturbation operator is well defined and gradients remain stable. For multiple contours, we compose regions with a simple pointwise max, and a smooth union is an interchangeable option. We will include the formulas in Appendix A1, as well as open sourcing the code, to make these choices easy to follow, as reproducibility and adaption is the main aim of the work presented here.
>
> **On runtime and practicality.** We report runtime under fixed hardware and protocol. Per image, Smooth Mask is about 6 s, our method about 8s with early stopping, while one-pass saliency runs in milliseconds. Memory is dominated by the backbone’s backward pass; contour parameters are negligible. We are working toward more efficient implementations, including mixed precision, smaller Fourier budgets on easier cases, batched backbone evaluations, and better initialization, but we consider it to be outside the scope of this introductory paper as the engineering practicalities may dilute the core idea of the method. Here is the comparison table:
>
> | **Method**               | **Runtime (s)**        |
> |---------------------------|------------------------|
> | Gradient SHAP             | 0.0318 ± 0.0025        |
> | Integrated Gradients      | 0.0986 ± 0.0055        |
> | Smooth Mask               | 6.0833 ± 0.0444        |
> | Grad-CAM++                | 0.0195 ± 0.0025        |
> | Extremal Contour (ours)   | 8.5740 ± 2.5312        |
>
>
>
> The contribution is a simple geometric parameterization that produces connected, readable regions with competitive faithfulness and robustness. The expressivity–simplicity and speed–faithfulness trade-offs are explicit rather than hidden. Multi-contour explanations and the area sweep offer practical flexibility when scenes are complex or when graded insight is needed. We are really grateful for this review as we believe it highlights where this approach is most useful and where it should be extended next.

---

### Official Review · Reviewer_3o42 · 2025-10-08
**Good paper, application scope is a bit limited**

**Rating:** 4
**Confidence:** 3

**Summary:**

The paper introduces a training-free, differentiable contour-based explanation method that replaces pixelwise perturbation masks with smooth, Fourier-parameterized contours optimized under the extremal perturbation objective. The authors identify a key limitation in existing perturbation-based explainability methods—fragmented or noisy dense masks—and address it by enforcing topological simplicity (single connected region) and parametric compactness. The method builds directly upon the established “extremal perturbation” framework [Fong et al. 2019], modifying it with a Fourier contour parameterization that guarantees differentiability and shape regularity by construction.

**Strengths:**

The experiments are well-structured and multi-faceted, covering both qualitative and quantitative evaluations on ImageNet and COCO datasets with supervised (ResNet-50) and self-supervised (DINO) models. This evaluation suite is comprehensive using established metrics such as Relevance Rank, Relevance Mass, entropy, Gini index, and perturbation-based faithfulness correlations. Quantitatively, the method achieves parity or improvement over baselines like Smooth Mask and Grad-CAM++, with particularly strong gains in relevance mass and robustness for self-supervised models. The paper is clear and well-written.

**Weaknesses:**

The method requires iterative optimization with a pretrained model in the loop. Although each contour has few parameters, computing embeddings and backpropagation per iteration introduces non-trivial cost relative to one-pass saliency or gradient-based methods. The paper acknowledges this but provides no runtime analysis.

Although the evaluation includes both supervised and self-supervised models, it is limited to classification networks (ResNet-50, ViT-B/16). It remains unclear how well the method generalizes to dense prediction tasks such as segmentation or detection.

By construction, the method outputs binary (region-based) explanations rather than graded saliency. While this aids interpretability, it limits fine-grained insights into which pixels within the contour are most responsible for the decision.

**Justification:**

The approach is technically sound, original, and clearly written, with careful experiments that validate its advantages. Despite minor limitations, the paper offers a valuable new direction in explainable AI by embedding geometric priors into perturbation-based attributions.

---

> ### Author Rebuttal · Authors · 2025-10-21
>
> Thank you for the thoughtful review and we greatly appreciate the positive comments. We address the three weaknesses raised in the review below.
>
> **Runtime.** We added a runtime section in the appendix, with a fixed hardware and protocol. As expected, our method is slower than one-pass saliency (about 8s per image) but similar in scale to other perturbation-based approaches (for example, Smooth Mask takes about 6s). Early stopping keeps runtimes usable. We also list straightforward engineering speedups, including feature caching, smaller Fourier budgets, and batched backbone calls. Since the optimization converges, a good initialization of the contour should reduce time further, and we plan to explore this in the future with a more application-focused objective.
>
> | **Method**               | **Runtime (s)**        |
> |---------------------------|------------------------|
> | Gradient SHAP             | 0.0318 ± 0.0025        |
> | Integrated Gradients      | 0.0986 ± 0.0055        |
> | Smooth Mask               | 6.0833 ± 0.0444        |
> | Grad-CAM++                | 0.0195 ± 0.0025        |
> | Extremal Contour (ours)   | 8.5740 ± 2.5312        |
>
>
> **Beyond classification.** The method we are presenting here is model-agnostic. For detection, the backbone still produces embeddings and target scores, so we can optimize the contour with respect to a chosen box or query score with unchanged gradients. For segmentation, the extension is more complex and would benefit from exploring which loss yields the most informative attribution. The method is simple by design, so users can choose losses that reflect their analysis goals. We will clarify this path in the discussion.
>
>
> **Binary vs. graded explanations.** We use a single, simply connected region by design. This avoids fragmented, noisy masks and provides a clear, topologically coherent summary when a crisp region is most useful. We agree that graded attribution is valuable. In Section 3.5 we introduce a fixed-area sweep that produces nested contours and an area–faithfulness curve, giving a hierarchical, graded view at the region level. Perturbation-based soft masks are closely related and emphasize region-level effects rather than per-pixel saliency. Our emphasis in this work is on topologically consistent regions with clear semantics.

---

### Meta-Review · Area_Chair_sDsg · 2025-10-31

**Recommendation:** Accept (Oral)
**Confidence:** 4

**Metareview:**

The idea of embedding geometric priors into perturbation-based attribution is interesting and novel. and it is worth sharing with the community for further discussion. The writing quality and clarity were repeatedly highlighted as positives by the reviewers.

The main concerns (the expressivity limits, lack of graded, pixel-level attribution, the computational cost, and the relatively narrow experimental scope) were effectively addressed in the rebuttal -- both using additional quantitative results and qualitative arguments (that the reviewers found defensible). The consensus is that this is a solid, well-motivated, and clearly presented contribution that opens a promising direction for geometrically constrained explainability.

---

### Decision · Program_Chairs · 2025-11-05

**Decision:**

Accept (Oral)

**Comment:**

We recommend an oral and a poster presentation given the AC and reviewers recommendations.